# Aramid Honeycomb Cores under Constant Pressure: Unveiling the Out-of-Plane Compression Deformation

**DOI:** 10.3390/polym16141974

**Published:** 2024-07-10

**Authors:** Xinzheng Huang, Xin Hu, Jinzhan Guo, Dechao Zhang, Shunming Yao, Lihua Zhan, Bolin Ma, Minghui Huang, Lihua Zhang

**Affiliations:** 1College of Mechanical and Electrical Engineering, Central South University, Changsha 410083, China; 13322995346@163.com (X.H.); yaoshunming@csu.edu.cn (S.Y.); meeh@csu.edu.cn (M.H.); 2Shenzhen Kuang-Chi Cutting-Edge Technology Co., Ltd., Shenzhen 518000, China; 3Light Alloys Research Institute, Central South University, Changsha 410083, China; ladihu4@gmail.com (X.H.); guruci@163.com (J.G.); 213801004@csu.edu.cn (D.Z.); zhanglihua@csu.edu.cn (L.Z.)

**Keywords:** aramid honeycomb core, out-of-plane compression deformation, moisture infiltration, viscoelasticity, Burgers model

## Abstract

The primary challenge during the secondary bonding process of full-height honeycomb sandwich structures is the aramid honeycomb core’s height shrinkage. This paper systematically investigated the height evolution behavior of the honeycomb core by using a creep testing machine. The results showed that the out-of-plane compression deformation curve of aramid honeycomb cores is mainly divided into three stages: the dehumidification stage, the pressurization stage and the creep stage. Under conditions of high temperature and pressure, height shrinkage was attributed to the dehydration caused by moisture infiltration, and the compression creep resulted from the slippage of polymer molecular chains. Dehydration shrinkage is stable, whereas compression creep reflects typical viscoelastic polymer characteristics. By employing the viscoelastic Burgers mechanical model and applying the nonlinear surface fitting method, the total height shrinkage deformation behavior of the aramid honeycomb core during the curing process can be accurately predicted by summing the above three stages. This research contributes valuable insights for the manufacturing process of honeycomb sandwich structures.

## 1. Introduction

Composite materials are renowned for their lightweight, high strength and corrosion resistance [1,2,3]. As a type of composite material, honeycomb sandwich structures primarily consist of top and bottom face sheets and a honeycomb core, and the aramid paper honeycomb core is commonly employed as the core material in this structure. The aramid paper honeycomb core is a bionic honeycomb core material made of aramid paper impregnated with phenolic resin. It has a series of excellent properties, such as its light weight, energy absorption, high specific strength/stiffness, corrosion resistance, radiation resistance, etc., and is widely used in various types of aerospace vehicles [4,5,6]. According to the type of aramid paper used, there are two common types of aramid paper honeycomb. Among them, those made of poly (m-phenylene isophthalamide) fiber paper are often referred to as NOMEX^®^ honeycombs, and those made of poly (p-phenylene terephthalamide) fiber paper are often referred to as KEVLAR^®^ honeycomb. KEVLAR^®^ honeycomb has more outstanding mechanical and thermal properties than NOMEX^®^ honeycomb. For advanced aircraft, using KEVLAR^®^ honeycomb can achieve a greater weight reduction target [7].

In the edge structures of control surfaces in fighter jets, small unmanned aerial vehicles and other aircraft, a full-height honeycomb integral sandwich construction with upper and lower skins is widely applied. Different from the method applied in the conventional honeycomb sandwich panel structure manufacturing process, honeycomb bonding can significantly decrease structural weight and enhance manufacturing efficiency. During the manufacturing processes of the full-height honeycomb sandwich structure, a secondary bonding process is indispensable. However, because of the high temperature and pressure during the bonding procedure, the honeycomb core experiences shrinkage in the height direction and, consequently, depression appears on the surface of the honeycomb structure. Therefore, clearing the mechanism of this shrinkage process is important for maintaining surface shape and keeping structure size accuracy.

Little research on the shrinkage and creep properties of aramid honeycomb cores under high temperature and pressure has been published, and most work addresses the fundamental mechanical properties of aramid honeycomb sandwich structures, such as tension [8], compression [9,10,11], shear [12], three-point bending [13] and impact properties [14]. For the creep properties of aramid honeycomb sandwich structures, the published literature often applies three-point bending creep tests. For example, this experimental approach was utilized by Du et al. [15] to investigate the creep behavior of Nomex aramid honeycomb sandwich structures under various stress and humidity conditions, and the time-dependent creep strain curve based on the Burgers model was fitted. Meanwhile, Ishak et al. [16] conducted the three-point bending creep performance of honeycomb sandwich structures at different spans, and applied Findley’s power law to simulate the viscoelastic response of the cross-arm. These aforementioned studies predominantly focused on the creep formability of honeycomb sandwich structures by conducting three-point bending at room temperature, and factors such as temperature and pressure were ignored.

Furthermore, there is also limited research on the compressive creep behavior of aramid honeycomb cores in the in-plane direction. Gibson and Ashby [17] calculated the overall elastic moduli of porous structures in both the in-plane and out-of-plane directions from a mechanical perspective based on material properties, and validated the in-plane viscoelastic and creep deformations by experiment data. However, the creep characteristics in the out-of-plane direction of the porous structures were not investigated. Lin and Huang [18] utilized a representative volume element (RVE) to analyze the in-plane creep characteristics of hexagonal honeycombs with non-uniform thickness cell edges. However, their investigation did not extend to examining the out-of-plane creep behavior, leaving a potential avenue for further research in this domain. Balkan and Mecitoglu [19] utilized a dynamic mechanical analyzer (DMA) to identify the viscoelasticity of aramid sandwich samples and carbon/epoxy resin samples. They investigated the nonlinear dynamic response of viscoelastic sandwich plates under explosion load. However, they did not explore the creep properties of aramid honeycomb cores, despite their viscoelastic characteristics. It can be found from the above works that the existing body of research has not fully revealed the creep characteristics of honeycomb structures in different directions.

Apart from the creep attribute under high pressure or stress, aramid honeycomb core experiences shrinkage deformation due to dehydration at high temperatures. The primary cause of this shrinkage is the hygroscopic expansion of the aramid fibers and phenolic resin that occurred before the test. There is a relatively larger body of research concerning the hygroscopicity of aramid honeycomb cores [20,21]. For example, Sala [22] investigated the infiltration levels and performance impacts of different liquids (water, fuel and dichloromethane) on aramid fibers and carbon fibers by utilizing an optical microscopy, and significantly demonstrated the swelling of aramid polyamide fibers after the wet treatment, confirming the moisture absorption swelling effect at the microstructural level. Guo et al. [23] provided a microscopic explanation for the moisture-induced expansion mechanism in composite materials, i.e., the formation of one or more hydrogen bonds between bound water and the molecular chains of fibers and resins.

Based on the observed out-of-plane shrinkage phenomenon under high temperature and pressure condition, this study systematically investigated the out-of-plane compressive creep behavior of aramid honeycomb cores. In this study, a creep test machine was employed to simulate the common forming process environment (temperature and pressure parameters), and the compressive creep phenomenon of aramid honeycomb cores was discussed in detail for the first time. After the conclusion of this experiment, the viscoelastic Burgers mechanical model was further applied to fit the time–compressive creep strain curve of the aramid honeycomb core in the creep stage. Then, the shrinkage deformation curve equation of the aramid honeycomb core was obtained by superimposing the deformations of the dehumidification, pressurization and creep stages. The research results of this paper have great guiding significance for the manufacturing process of aramid honeycomb sandwich structure.

## 2. Materials and Methods

### 2.1. Experimental Materials

Supplied by Shenzhen Kuang-chi Cutting-edge Technology Co., Ltd. (Shenzhen, China), aramid honeycomb core with a density of 48 kg/m^3^ was utilized in this work. The aramid honeycomb core has a repetitive hexagonal unit structure, and different numbers of complete honeycomb lattices may have appeared under the same sample size. To ensure consistency, the W direction of the sample included 9 complete honeycomb lattices, and the L direction included 11 complete honeycomb lattices, for a total of 99 complete honeycomb cores. The specific sample dimensions and honeycomb lattice dimensions are illustrated in Figure 1.

### 2.2. Experimental Methods

This experiment was conducted using the SUST creep testing machine, which has a maximum control temperature of 350 °C and a force accuracy of 0.01 N. In the experiment, extensometers manufactured by Weihai Dikong Electronic Technology Co., Ltd. (Weihai, China), with the model number D-KB10-05-T4 and a displacement accuracy of 0.0001 mm, were employed. To adapt to the needs of out-of-plane compression creep, the experimental fixture was self-modified, as shown in Figure 2. In the mechanism of the out-of-plane compression creep instrument, the plane compression platens primarily convert the pressure provided by the creep testing machine into a planar pressure. Extension rods were employed to transmit the deformation of the sample height to the extensometers. Simultaneously, the environmental chamber controlled the ambient temperature, while extensometers recorded the sample’s height shrinkage in real-time. The specific working mechanism of the apparatus is illustrated in Figure 3.

In this study, the selected temperature conditions were chosen as they are commonly used in honeycomb structure manufacturing processes: a medium curing temperature of 120 °C and a high curing temperature of 180 °C, with set pressure conditions of 0.2 MPa and 0.3 MPa, respectively. Each experimental condition employed five identical aramid honeycomb cores of the same material, with the same dimensions and lattice counts. Initially, a force of 10 N was applied to the specimen to ensure intimate contact between the plane compression platens and the surfaces of the aramid honeycomb core, thereby enhancing measurement accuracy. Upon reaching the processing temperature, no additional pressure was directly applied; instead, the specimen was maintained at this temperature for 30 min to eliminate moisture from within the aramid honeycomb core and mitigate the influence of humidity on its creep behavior. In industrial settings, subjecting the honeycomb core to temperatures exceeding 100 °C for 30 min effectively removes internal moisture. Consequently, whether baked at 120 °C or 180 °C for 30 min, water molecules within the honeycomb core are efficiently expelled. It is worth noting that the selected test samples have a buckling strength of 1 MPa. Therefore, under the experimental pressure conditions (0.2 MPa, 0.3 MPa), no structural buckling in the aramid honeycomb cores occurred. The specific experimental procedures are illustrated in Figure 4.

## 3. Results

Under each experimental condition, a dataset exhibiting relative stability and minimal fluctuations was selected. Subsequently, sampling was applied at every fifth data point to plot the corresponding time–compression deformation curves, as depicted in Figure 5. It can be observed that the aramid honeycomb cores exhibited a consistent trend during the out-of-plane compression creep test, characterized by compression–expansion–platform–pressurization creep.

Based on this trend, a typical results trend curve was plotted, which delineated the consistent variation pattern of the honeycomb core’s height direction under high temperature and pressure condition, as shown in Figure 6. Among them, point O represents the initial point of the experiment, point A is the first sampling point beyond 10 s, point B indicates the peak shrinkage of the aramid honeycomb core, point C is the expansion peak, point D denotes the end of the dehumidification stage and the beginning of the pressurization stage, and point E marks the end of the pressurization stage and the beginning of the creep stage. In Figure 6, three distinct deformation stages are observed: the OD stage, the DE stage and the EG stage. The OD stage is primarily attributed to the volume contraction caused by the dehydration of the honeycomb at high temperatures, thus named the dehumidification stage. The DE stage exhibited obvious elastic characteristics, representing the elastic deformation of the honeycomb under out-of-plane compression, named the pressurization stage. The deformation observed in the EG stage is the deformation attained under the conditions of the creep test, and therefore, the EG stage is designated as the creep stage.

Define total deformation Δ*L*, where Δ*L =* Δ*L*_1_
*+* Δ*L*_2_
*+* Δ*L*_3_ and Δ*L*_1_
*=* Δ*L*_11_
*+* Δ*L*_12_. The deformation Δ*L*_1_ during the dehumidification stage mainly consists of two parts: the elastic deformation Δ*L*_11_ in the OA stage and the humidity deformation Δ*L*_12_ in the AD stage. The deformations during the pressurization and creep stages are represented by Δ*L*_2_ and Δ*L*_3_, respectively.

### 3.1. Dehumidification Stage

The deformation Δ*L*_1_ in the dehumidification stage is mainly composed of two parts: the elastic deformation Δ*L*_11_ in the OA stage and the humidity deformation Δ*L*_12_ in the AD stage. For each experimental condition with five data sets, the highest and lowest values were excluded, and then the three central data sets for Δ*L*_11_ and Δ*L*_12_ were selected to be presented in Table 1.

Δ*L*_11_ represents the deformation resulting from the application of a 10 N force at room temperature, while Δ*L*_12_ is attributed to moisture dehydration. Given the consistent experimental site and a dehumidification temperature exceeding 100 °C, it is theoretically expected that Δ*L*_11_ and Δ*L*_12_ under the four experimental conditions would be fixed at specific values. Consequently, the average values of all Δ*L*_11_ and Δ*L*_12_ were calculated separately and are presented in Table 1. The errors in Δ*L*_11_ and Δ*L*_12_ arise from the discrepancies in the experiment’s starting time.

To investigate whether the deformation Δ*L*_12_ of the 20 mm aramid honeycomb core remains consistently around 0.058 mm during the dehumidification stage, two sets of aramid honeycomb cores with identical dimensions were measured for height in the T direction before and after drying, as shown in Table 2. By summing up the height differences of the four points before and after drying and taking the average, the moisture-induced deformation of the honeycomb was measured to be 0.055 mm, which closely matches the data obtained from the creep testing machine.

Based on a detailed observation of Figure 6, it is further revealed that the aramid honeycomb core does not remain in a continuous state of contraction during the dehumidification phase. Specifically, after reaching the shrinkage peak point B, the aramid honeycomb core exhibits an unexpected expansion trend (as seen in the BD segment of Figure 6). That is because, prior to the initiation of the experiment, hydrogen bonds had formed between the water molecules and the polar groups within the aromatic polyamide fiber/phenolic resin, consequently resulting in the expansion of the aramid honeycomb core [23,24]. In order to confirm the hygroscopic expansion of the aramid honeycomb samples prior to experimentation, this study utilized a Thermo Fisher Nicolet iS20 infrared spectrometer to measure the functional groups of pure aramid paper before and after drying. As shown in Figure 7, the infrared spectrum revealed hydroxyl groups exhibiting stretching vibrations around 3300–3400 cm^−1^ and bending vibrations around 1500 cm^−1^. These characteristic peaks indicate that prior to the experiment, the aramid fibers had already formed hydrogen bonds with external water molecules, thereby also validating the mechanism that aramid honeycomb structures expand due to moisture absorption under humid conditions.

Since the aramid honeycomb core had already experienced hygroscopic swelling before the experiment began, at the initial stages of the experiment, the increase in environmental temperature disrupted the hydrogen bonds formed between water and the polar groups, causing water evaporation and resulting in a contraction phenomenon (as seen in the AB segment of Figure 6). As the temperature further increased to the conditioned level, the enhanced thermal motion of molecules weakened the intermolecular interactions, leading to an increase in the polymer’s free volume and resulting in the overall expansion of the aramid honeycomb core (as seen in the BC segment of Figure 6). After the expansion phenomenon and before the pressurization stage, there was a relatively long duration of plateau (as seen in the CD segment of Figure 6). This plateau stage marks the end of the dehumidification stage and signifies the complete removal of moisture from the aramid honeycomb core.

### 3.2. Pressurization Stage

The averages of Δ*L*_2_ and Δ*L*_3_ under each experimental condition are presented in Table 3. According to Table 3, it can be observed that the deformation Δ*L*_2_ during the pressurization stage occupies a significant proportion of the total deformation Δ*L*_2_, approximately around 55%. The magnitude of Δ*L*_2_ is primarily influenced by the applied force, meaning that as the load increases, the corresponding Δ*L*_2_ also increases. It can also be found that under the condition of keeping the temperature constant, when the pressure is expanded to 1.5 times, the deformation Δ*L*_2_ does not increase in the same proportion, revealing a nonlinear relationship. It shows that under high temperature and pressure conditions, the aramid paper honeycomb core impregnated with phenolic resin is not a pure linear elastomer, nor is it a pure viscous fluid, and it is more likely to be a viscoelastic material between elasticity and viscosity. It is worth noting that this finding is consistent with the results obtained by Balkan and Mecitoglu using a dynamic mechanical analyzer (DMA) to measure the viscoelasticity of aramid honeycomb cores [19].

### 3.3. Creep Stage

#### 3.3.1. Explanation of Creep Stage

Aramid fibers belong to the category of aromatic polyamide organic polymers, characterized by their molecular structure consisting of rigid main chains aligned along the fiber axis, composed of aromatic rings and amide bonds. These main chains interact through strong hydrogen bonding, collectively forming a rod-like fiber structure [25,26]. The manufacturing process of aramid fibers employs the dry-jet wet spinning technique, yielding thermoplastic polymer fibers with a distinct skin-core structure. Within this structure, the molecular chains at the fiber’s core exhibit segments that are not parallel to the fiber axis, which leads to a certain degree of creep behavior under stress. Additionally, under experimental conditions, an increase in temperature causes the softening of the phenolic resin matrix, increasing the likelihood of fiber slippage within the resin, ultimately resulting in the creep phenomenon observed in aramid honeycomb structures.

It is evident from Table 3 that the aramid honeycomb core manifests creep phenomena. The magnitude of creep deformation, denoted as Δ*L*_3_, constitutes a minor fraction, not surpassing 10% of the total deformation. However, Δ*L*_3_ is positively correlated with temperature, time and pressure. With the increase in temperature, time and pressure, the deformation of Δ*L*_3_ is more significant. The experimental results indicate that the honeycomb composite material (consisting of thermoplastic poly (p-phenylene terephthalamide) fibers and thermosetting phenolic resin) exhibits the creep phenomenon characteristic of viscoelastic materials under high temperature and pressure. The heights of the specimens were measured before and after the experiment (once moisture equilibrium was achieved), indicating that the plastic deformation and the deformation attributed to creep (Δ*L*_3_) of the honeycomb were found to be nearly consistent.

#### 3.3.2. Burgers Model Introduction and Fitting Results

Under the influence of specific temperature and small constant stress, viscoelastic polymer will exhibit the creep phenomenon. In order to characterize the viscoelastic properties of polymers, researchers often use simple mechanical elements such as Hooke’s spring and Newton’s dashpot. In the related research, mechanical models capable of describing the creep phenomenon of polymeric materials have been developed through series, parallel and other combinations of Hooke’s spring and Newton’s dashpot [27]. Since the creep deformation of the viscoelastic honeycomb core during the experiment depends simultaneously on time, temperature and pressure, its creep deformation is fitted individually. Furthermore, since the four-element Burgers model can encompass various scenarios described by both the two-element (Maxwell model and Kelvin model) and three-element models, Burgers model was directly chosen to fit the time–creep strain curves under each experimental temperature condition. The Burgers model’s constitutive equation and model are presented in Equation (1) and depicted in Figure 8, respectively.

The Burgers constitutive equation is as follows:(1)$$\varepsilon (t)=\frac{\sigma_{0}}{E_{1}}+\frac{\sigma_{0}}{\eta_{1}}t+\frac{\sigma_{0}}{E_{2}}(1-\;e\mathrm{xp}(-E_{2}t/\eta_{2}))$$
where *ε*(*t*) represents the strain; *σ*_0_ indicates the initial stress; *E*_1_ and *E*_2_ denote the elastic modulus of the left and right springs, respectively; and *η*_1_ and *η*_2_ are the viscosity of the left and right dashpot, respectively.

The creep experimental data of aramid honeycomb under different experimental conditions are shown in Figure 9. Through the analysis of the experimental conditions and the creep strain, it was found that an increase in temperature or pressure leads to the increase in creep amount. Moreover, an increase in pressure delayed the time it took for the aramid honeycomb core to reach a steady creep state. Under the experimental conditions of 120 °C, 0.2 MPa and 180 °C, 0.3 MPa, the honeycomb did not show a distinct steady creep state. This could potentially be attributed to the damage sustained by the aramid fiber/phenolic resin interface or the continuous molecular chain slippage of the aramid fiber under high temperatures and pressure. Subsequently, MATLAB R2020a software was employed to fit the creep experimental data at 120 °C and 180 °C using the nonlinear surface fitting method, treating time and stress as independent variables. The fitting parameters and errors of the Burgers model are presented in Table 4. Ultimately, the Burgers’ constitutive equations of creep strain at 120 °C and 180 °C are given by Equations (2) and (3), respectively.
(2)$$\varepsilon (t)=\frac{\sigma_{0}}{4.616}+\frac{\sigma_{0}}{25580}t+\frac{\sigma_{0}}{3.571}(1-e\mathrm{xp}(-3.571t/484.1))$$
(3)$$\varepsilon (t)=\frac{\sigma_{0}}{3.481}+\frac{\sigma_{0}}{8231}t+\frac{\sigma_{0}}{1.862}(1-e\mathrm{xp}(-1.862t/229))$$

According to the Burgers constitutive equation, it is evident that as *t*→0, the strain *ε*(*t*) approaches $\frac{\sigma_{0}}{E_{1}}$. Simultaneously, as *t→∞*, the strain *ε*(*t*) approaches $\frac{\sigma_{0}}{E_{1}}+\frac{\sigma_{0}}{\eta_{1}}t+\frac{\sigma_{0}}{E_{2}}$, and the strain rate $\dot{\varepsilon (t)}$ approaches $\frac{\sigma_{0}}{\eta_{1}}$. Furthermore, based on Equations (2) and (3), it was observed that under consistent nominal stress, at *t→*0, higher temperatures correspond to smaller *E*_1_, indicating greater creep deformation at elevated temperatures. At *t→∞*, higher temperatures correspond to smaller *η*_1_, indicating higher creep deformation rates under elevated temperatures. In addition to the above, *E*_1_, *E*_2_ and *η*_1_ collectively dictate the long-term creep deformation *ε*(*∞*), whereas *E*_2_ and *η*_2_ jointly influence the time required to achieve stable creep deformation.

It can also be observed from Table 4 that the application of nonlinear surface fitting methods to fit the Burgers model with creep test data yields a relatively ideal fitting effect. Furthermore, the fitting results exhibit good representativeness and are capable of predicting the creep curves under different stress levels at the same temperature conditions, which holds significant practical implications for engineering applications.

## 4. Conclusions

As the most widely used sandwich material in aerospace applications, the deformation of aramid honeycomb cores in the secondary bonding process has a great influence on the quality of the manufacturing process. In this paper, the compression phenomenon of para-aramid honeycomb cores under high temperatures and high pressure was systematically discussed. The main conclusions are as follows:The out-of-plane compression performance of the aramid honeycomb core was systematically investigated for the first time. Based on the compression deformation curve, the compression behavior can be divided into three stages: the dehumidification stage, the pressurization stage and the creep stage. These stages contribute approximately 35%, 55% and 10%, respectively, to the total deformation.During the dehumidification stage, the aramid honeycomb core undergoes a process of compression, expansion and plateau. The primary reason for the phenomenon of honeycomb cores is the combined effect of thermal expansion and dehydration shrinkage on aramid honeycomb cores. Prior to the experiment, hydrogen bonds formed between water molecules and the polar groups in the aramid honeycomb core, causing expansion. As the temperature increased, the hydrogen bonds were disrupted, leading to water evaporation and the contraction of the honeycomb core. Further temperature increases weakened intermolecular interactions, increasing the polymer’s free volume and resulting in the overall expansion of the aramid honeycomb core.In the pressurization stage, the deformation Δ*L*_2_ and pressure exhibit a nonlinear relationship, indicating that the aramid honeycomb core impregnated with phenolic resin is a not purely linear elastomer. Simultaneously, in the creep stage, the aramid honeycomb core exhibits compression creep phenomena related to time, temperature and pressure under high temperature and pressure conditions. Taking into account the aforementioned phenomena, the aramid honeycomb core under high temperature and pressure conditions can be characterized as a viscoelastic material.After the out-of-plane compression creep experiment, the nonlinear surface fitting method was employed to fit the creep deformation equations at various stress levels under a constant temperature, resulting in a high degree of fitting accuracy. This method is also capable of predicting the creep deformation behavior under the influence of varying stress levels. Finally, the total height deformation of the aramid honeycomb core under typical manufacturing processes can be obtained by integrating the dehumidification (Δ*L*_1_), pressurization (Δ*L*_2_) and creep (Δ*L*_3_) stages.

## Figures and Tables

**Figure 1 polymers-16-01974-f001:**
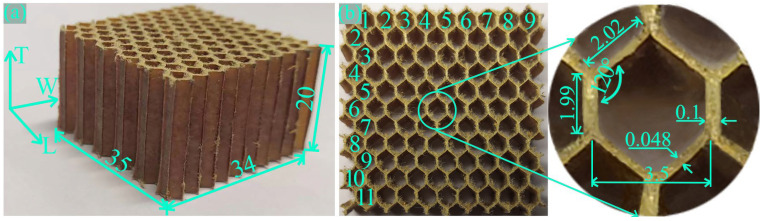
Detailed dimensions of experimental samples and honeycomb lattice structure. (**a**) Sample dimensions and (**b**) honeycomb lattice dimensions.

**Figure 2 polymers-16-01974-f002:**
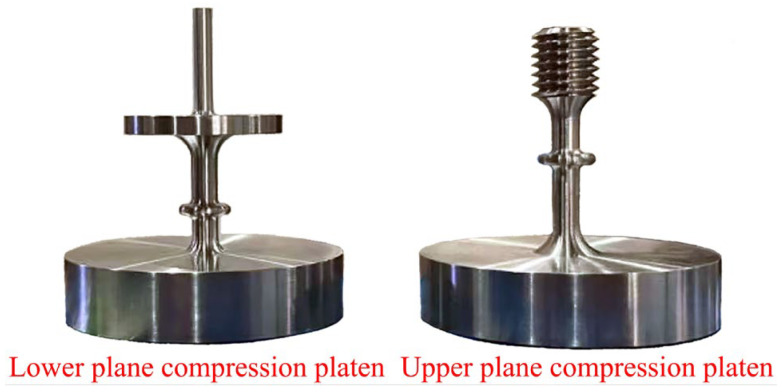
Upper and lower plane compression platens.

**Figure 3 polymers-16-01974-f003:**
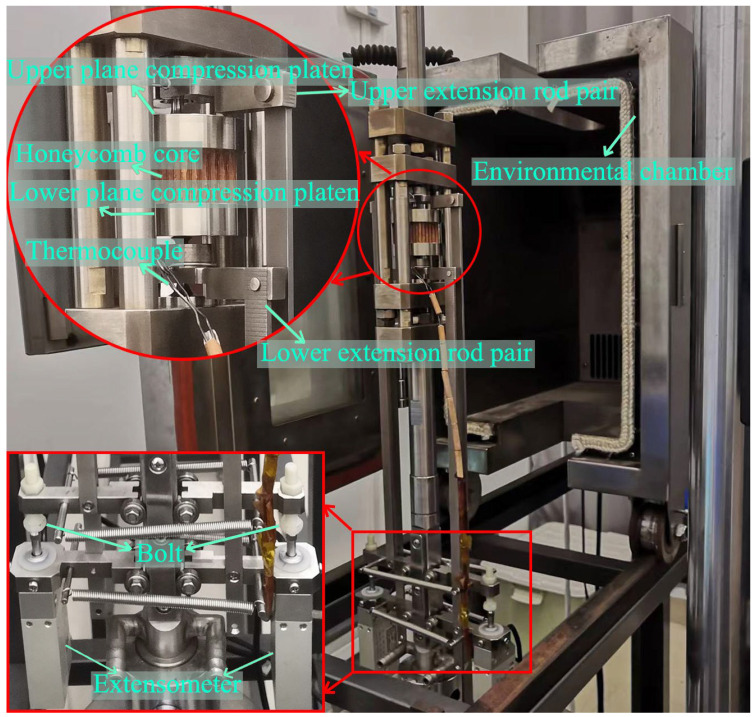
Out-of-plane compression creep experimental instrument.

**Figure 4 polymers-16-01974-f004:**
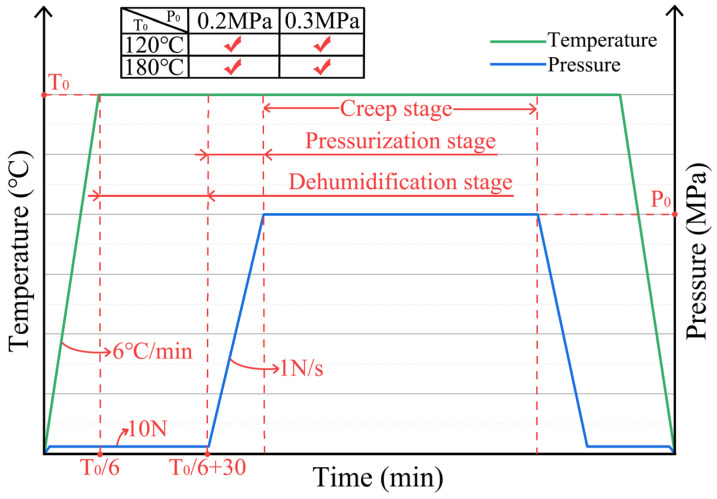
Specific experimental procedures.

**Figure 5 polymers-16-01974-f005:**
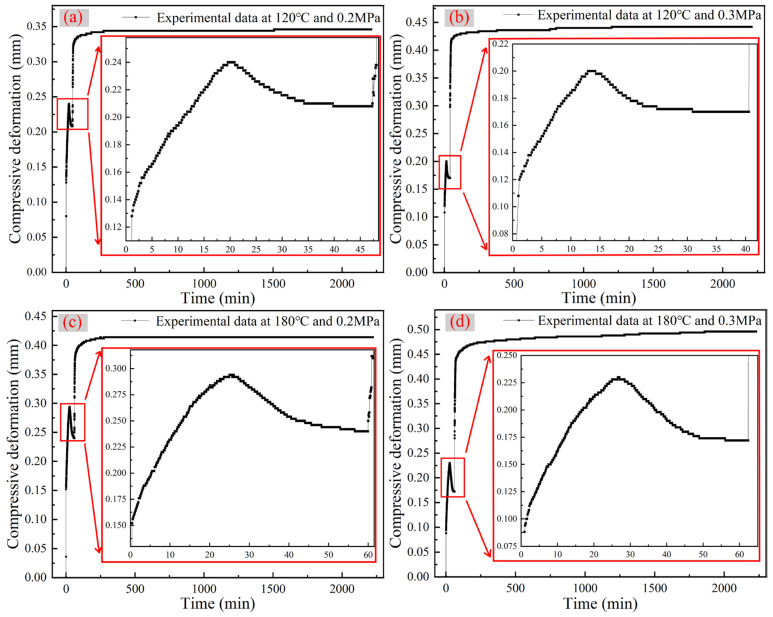
Time–compression deformation curves under various experimental conditions: (**a**) experimental data at 120 °C and 0.2 MPa, (**b**) experimental data at 120 °C and 0.3 MPa, (**c**) experimental data at 180 °C and 0.2 Mpa and (**d**) experimental data at 180 °C and 0.3 MPa.

**Figure 6 polymers-16-01974-f006:**
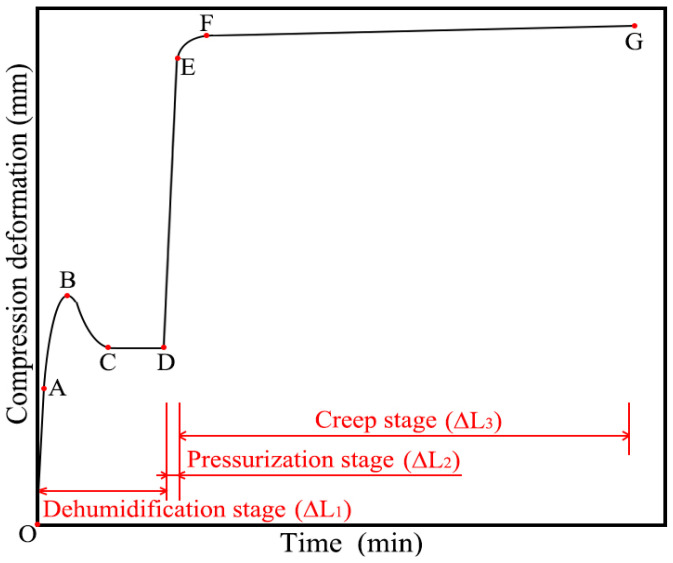
Typical trend curve of creep experiment.

**Figure 7 polymers-16-01974-f007:**
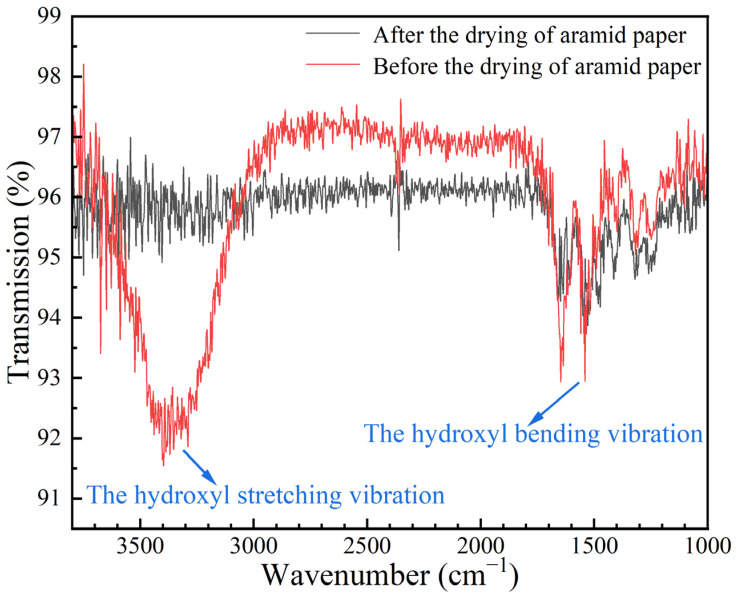
The infrared spectrum of aramid paper before and after drying.

**Figure 8 polymers-16-01974-f008:**
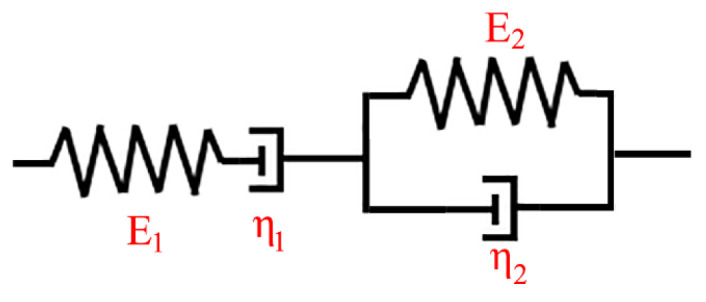
Burgers mechanical constitutive model.

**Figure 9 polymers-16-01974-f009:**
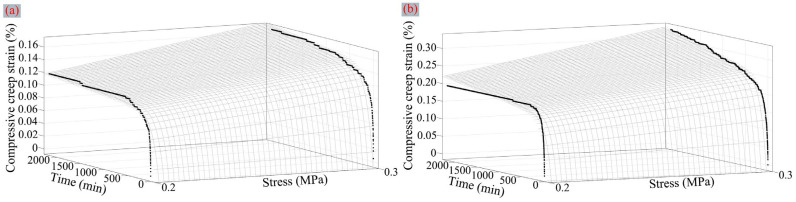
Time–creep strain curves and fitting results under various experimental conditions: (**a**) the creep curves and nonlinear surface fitting result at 120 °C and (**b**) the creep curves and nonlinear surface fitting result at 180 °C.

**Table 1 polymers-16-01974-t001:** Deformations of Δ*L*_11_ and Δ*L*_12_ under each experimental condition.

Experimental Conditions	Δ*L*_11_ (mm)	Δ*L*_12_ (mm)
120 °C, 0.2 MPa	0.112, 0.118, 0.132	0.060, 0.062, 0.066
120 °C, 0.3 MPa	0.088, 0.104, 0.110	0.046, 0.050, 0.058
180 °C, 0.2 MPa	0.100, 0.100, 0.130	0.050, 0.058, 0.066
180 °C, 0.3 MPa	0.106, 0.106, 0.116	0.054, 0.064, 0.064
Overall mean value	0.110	0.058

**Table 2 polymers-16-01974-t002:** Height of aramid honeycomb core before and after drying.

Sample	Before Drying		After Drying	
	Height 1 (mm)	Height 2 (mm)	Height 1 (mm)	Height 2 (mm)
Sample 1	19.95	19.83	19.89	19.77
Sample 2	20.00	20.02	19.95	19.97

**Table 3 polymers-16-01974-t003:** Deformation of each stage.

Experimental Conditions	Δ*L* (mm)	Δ*L*_1_ (mm)		Δ*L*_2_ (mm)	Δ*L*_3_ (mm)
		Δ*L*_11_ (mm)	Δ*L*_12_ (mm)		
120 °C, 0.2 MPa	0.392	0.110	0.058	0.198	0.026
120 °C, 0.3 MPa	0.442			0.242	0.032
180 °C, 0.2 MPa	0.430			0.224	0.038
180 °C, 0.3 MPa	0.492			0.260	0.064

**Table 4 polymers-16-01974-t004:** Summary of the fitted parameters for creep strain tested under experimental conditions based on the Burgers model.

Parameters	Experimental Conditions			
	120 °C	Error Range	180 °C	Error Range
$E_{1}$	4.616	(−0.088, +0.087)	3.481	(−0.038, +0.039)
$\eta_{1}$	25,580	(−800, +800)	8231	(−59, +58)
$E_{2}$	3.571	(−0.051, +0.051)	1.862	(−0.011, +0.01)
$\eta_{2}$	484.1	(−17.2, +17.2)	229	(−3.3, +3.2)
Fitting degrees	0.9226		0.9428	

## Data Availability

The original contributions presented in the study are included in the article, further inquiries can be directed to the corresponding author.

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
