# Peer review of "Aramid Honeycomb Cores under Constant Pressure: Unveiling the Out-of-Plane Compression Deformation"

_polymers, 2024, doi:10.3390/polym16141974_

Round 1

Reviewer 1 Report

Comments and Suggestions for Authors

The work investigates the height reduction of aramid honeycomb cores during the curing process in the manufacturing of honeycomb sandwich structures. Using a creep testing machine, the study identifies three main stages of out-of-plane compression deformation: dehumidification, pressurization, and creep. The height shrinkage is primarily caused by dehydration from moisture and compression creep due to the slippage of polymer molecular chains. By applying the viscoelastic Burgers mechanical model and Orthogonal Distance Regression (ODR), the study accurately predicts the total height reduction, improving the understanding and control of this process in manufacturing. I think a major revision is necessary before the publication of this manuscript. Overall, it could be considered for publication in Polymers after addressing the following;

1.     What are the key innovative aspects of your study compared to previous research on the height reduction of aramid honeycomb cores during curing?

2.     How does your application of the viscoelastic Burgers mechanical model and Orthogonal Distance Regression (ODR) improve the prediction accuracy of height shrinkage compared to traditional methods?

3.     In what ways does your study enhance the practical understanding of compression creep and dehydration-induced shrinkage in aramid honeycomb cores under high temperature and pressure conditions?

4.     What potential applications or implications does your research have for the manufacturing process of honeycomb sandwich structures that were not addressed in earlier studies?

5.     In the Results section, explain the choice of the typical data set in more detail. Clarify how the representative curve was selected and its significance.

6.     Provide a more detailed explanation of the mathematical model used, particularly how the Burgers model parameters were determined and validated.

7.     Suggest potential future research directions based on the findings.

Comments on the Quality of English Language

Please see the comments

Reviewer 2 Report

Comments and Suggestions for Authors

Paper is well written and few editing issues have to be solved. They are all mentioned in the attached file.

Mainly, leave space between number and units. Remake the font size of dimensions and color in Fig. 1. Change font color in Fig. 3 as to be seen clearly, Compression platens and not compression molds.

Reviewer 3 Report

Comments and Suggestions for Authors

The manuscript is devoted to the study of the physical and mechanical properties of aramid honeycomb cores at various temperatures and pressures. Overall, this is an interesting work that contains important information. But there are a number of questions regarding the processing and analysis of the results obtained:

1. The choice of experimental temperatures is explained a little, but what determines the choice of pressure?

2. The caption for Figure 1 seems a bit meaningless.

3. The red signatures in Figure 3 are difficult to see.

4. Figure 6 represents a typical trend curve of a creep experiment and contains a number of important points. However, the description of these points does not contain sufficient information. For example, it is not clear how the position of point A was determined? In Fig. 5 clearly shows that at the initial stage of the curve there is actually a large jump between the points. Usually, a tangent is drawn at this place and the module is determined from it (by the way, for some reason the authors did not do this). But according to the authors’ model, it does not seem that point A corresponds to the departure of the curve from the tangent (and the determination error here will be very high). Nevertheless, at the initial stage, the authors pay great attention to point A, since they divide the OD section into OA and AD (deformation L11 and L12, respectively). If we evaluate the numerical values of L11 and L12 in Table 1, then they decrease with increasing both temperature and pressure. But if we evaluate the position of point B in Fig. 5, then the influence of pressure will be the same, and the influence of temperature will be opposite (compare Fig. 5 a and c). What is the error in determining the position of these points and can we then talk about the tendency of the influence of temperature and pressure on them at this stage?

5. Line 180. If the authors describe hydrogen bonds, then it would be logical to provide supporting data from IR spectroscopy. In the introduction they referred to an article that described this, but at this point in the text it is mentioned without any evidence.

6. Lines 220-221. The authors write that L3 correlates with increasing temperature and pressure. In general, according to the numerical data in Table 3, this is true. But if we evaluate the L3 values from Figure 5, then it is difficult to talk about this unambiguously. At higher pressures, there is a tendency for a longer time to reach the plateau (section FG), and the interval of the main inflection (section EF) is approximately the same. But despite the fact that in Figure 6 there is a division into two sections, when analyzing the numerical values of L3, the authors evaluate only the overall change. In general, the authors do not describe reaching an equilibrium state at all when describing this stage of creep.

7. In the last part of the article, the authors made an attempt to correlate the obtained experimental curves with the calculated model. In fact, to form a model, you need a clear analysis of physical phenomena in all sections of the deformation curve! Nevertheless, the presented numerical calculations seem to agree well with the experimental data. But there is a lack of detailed explanations of exactly how the coefficients of the equations were chosen in Table 4 and 5. In what order was this done? What is the error in determining these coefficients? Are the resulting equations universal and carry some kind of physical meaning, or is it an adjustment to a couple of specific curves? Why were only the creep areas assessed separately? All this needs to be explained in the text.

Round 2

Reviewer 1 Report

Comments and Suggestions for Authors

this manuscript is ready now for publication.

Reviewer 3 Report

Comments and Suggestions for Authors

The authors generally answered all questions. It’s a bit of a pity that the comments in the replies are much more interesting and detailed than the changes in the text.